# Phytochemical Screening by LC-ESI-MS/MS and Effect of the Ethyl Acetate Fraction from Leaves and Stems of *Jatropha macrantha* Müll Arg. on Ketamine-Induced Erectile Dysfunction in Rats

**DOI:** 10.3390/molecules27010115

**Published:** 2021-12-25

**Authors:** Johnny Aldo Tinco-Jayo, Enrique Javier Aguilar-Felices, Edwin Carlos Enciso-Roca, Jorge Luis Arroyo-Acevedo, Oscar Herrera-Calderon

**Affiliations:** 1Department of Human Medicine, Faculty of Health Sciences, Universidad Nacional de San Cristobal de Huamanga, Portal Independencia 57, Ayacucho 05003, Peru; johnny.tinco@unsch.edu.pe (J.A.T.-J.); enrique.aguilar@unsch.edu.pe (E.J.A.-F.); edwin.enciso@unsch.edu.pe (E.C.E.-R.); 2Department of Dynamic Sciences, Faculty of Medicine, Universidad Nacional Mayor de San Marcos, Av. Miguel Grau 755, Lima 15001, Peru; jarroyoa@unmsm.edu.pe; 3Department of Pharmacology, Bromatology and Toxicology, Faculty of Pharmacy and Biochemistry, Universidad Nacional Mayor de San Marcos, Jr. Puno 1002, Lima 15001, Peru

**Keywords:** antioxidant agent, aphrodisiac, liquid chromatography, huanarpo, sexual behavior, experimental model, LC-MS plant extract

## Abstract

*Jatropha macrantha* Müll Arg. L is also known as “huanarpo macho” and used in the Peruvian traditional medicine as an aphrodisiac and erectile dysfunction (ED). The aim of this study was to determine the phytochemical constituents in leaves and stems ethyl acetate fraction (LEAF and SEAF) of *J. macrantha* and to compare the antioxidant activity and the ameliorative effect on ketamine-induced erectile dysfunction in rats. The phytochemical constituents were determined by LC-ESI-MS/MS, the total phenolic compounds and total flavonoids (TPC and TF) by Folin-Ciocalteu and aluminum chloride, respectively. The antioxidant activity was determined by DPPH, ABTS, and FRAP assays. Experimental groups were divided as follows: I: negative control; II: positive control (ketamine at 50 mg/ kg/d); III: sildenafil 5 mg/kg; IV, V, VI: LEAF at 25, 50 and 100 mg/kg, respectively, and VII, VIII, IX: SEAF at 25, 50, and 100 mg/kg, respectively. The phytochemical analysis revealed the presence mainly of coumarins, flavonoids, phenolic acids, and terpenes. TPC of LEAF and SEAF were 359 ± 5.21 mg GAE/g and 306 ± 1.93 mg GAE/g, respectively; TF in LEAF and SEAF were 23.7 ± 0.80 mg EQ/g, and 101 ± 1.42 mg EQ/g, respectively. The DPPH, ABTS, FRAP in SEAF were 647 ± 3.27; 668 ± 2.30; and 575 ± 2.86 μmol TE/g, respectively, whilst LEAF showed 796 ± 3.15; 679 ± 0.85; and 806 ± 3.42 μmol TE/g, respectively. Regarding sexual behavior, LEAF showed a better effect in mount frequency, intromission frequency, ejaculation frequency, mount latency, intromission latency, ejaculatory latency, and post ejaculatory latency than SEAF. As conclusion, LEAF of *J. macrantha* at 50 mg/kg showed a better effect on sexual behavior in male rats with erectile dysfunction than SEAF but not higher than sildenafil.

## 1. Introduction

Erectile dysfunction (ED) is a physiological or pathological condition characterized by the disability to achieve or maintain a penile erection during a sexual activity in men and nowadays is considered the second most frequent problem of sexual dysfunction in men [1]. In Europe the prevalence was of 52% in men aged 40–70 years [2] as well as estimated projections of prevalence around 322 million by 2025 in the world [3]. ED is a common condition associated with older adults and significantly compromises the sexual performance, personal satisfaction, commitment and low self-esteem [4]. On the other hand, exist many conditions and factors linked to ED, generally associated with comorbidities such as diabetes, hypertension, stress, cardiovascular diseases, depression, anxiety, mood, and lower urinary tract disorders. Furthermore, it is known that some risk factor could exacerbate ED, among obesity, metabolic syndrome, alcohol, and smoke [5]. Additionally, the main predominant neurotransmitter related to penile erection is nitric oxide (NO) released from noncholinergic fibers and the endothelium [6]. According to the guidelines of the American Urological Association and the European Urology Association, the management of ED as first strategy involves lifestyle modifications and second, the use of the phosphodiesterase type 5 inhibitor such as sildenafil [7]. However, new therapies such as platelet-rich plasma and stem cells could be promissory options in the future [8,9]. In Latin America three commercial drugs are available as main treatment such as sildenafil (Viagra), tadalafil (Cialis), and vardenafil (Levitra) [10]. 

The Euphorbiaceae family is represented by about 8100 species and 300 genera; they are distributed in tropical and subtropical regions [11]. Within this family, several useful species particularly the genera Croton, Euphorbia, and Jatropha are used as medicinal plants Currently, more than 80% of Jatropha genus are used in folk medicine from Afrika, Asia and Latin America [12]. Moreover, this genus has more than 175 species and are recognized as important sources of secondary metabolites with a broad spectrum of biological functions [13]. Extracts obtained with different solvents and isolated compounds from species of Jatropha genus are known by its biological activities reported in the literature such as antioxidant, anti-inflammatory, insecticidal, larvicidal, AChE inhibitory, cytotoxicity, antimicrobial, antiviral, antifungal, and aphrodisiac. Some species have demonstrated biological activities such as *J. curcas* (antibacterial, antioxidant, insecticide, cytotoxic, *J. gossypiifolia* (antifertility, anticancer, antiinflammatory, anticoagulant), *J. tanjorensis* (anticancer, antimicrobial, and antioxidant), *J. multifida* (ethyl acetate: antiviral and anticancer), *J. dioica* (antioxidant) and among others [14]. Within its secondary metabolites have been identified peptides, alkaloids, lignans, flavonoids, phenolic acids, coumarins, and mainly terpenes [15]. 

*Jatropha macrantha* Müll. Arg. is also known as “huanarpo macho” in Peru and considered a medicinal plant potentially as aphrodisiac [16]. Other studies refer a bronchodilator effect [17], antimelanogenic, antiinflammatory [18], inhibitor of nuclear factor–ĸB and hypoxia inducible factor 1 (HIF-1) in tumor cells [19], modulatory of estradiol, progesterone and testosterone [20], antioxidant and inhibitory of aldose reductase [21]. Otherwise, a new tendency in the consumption of natural products, which includes dietary supplements, herbal medicines or medicinal plants are increasingly for treatment of ED, particularly sold as over-the-counter products [22]. Due to lack of therapeutical options to revert ED, the inclusion of *J. macrantha* as a nature alternative treatment might be a great opportunity to focus on the pharmacological investigation to this promissory specie from Peruvian andes. In this study was used stems and leaves of *J. macrantha*, which were fractioned to obtain the ethyl acetate fraction to obtain a rich extract mainly in phenolic compounds and avoiding the interfering of waxes and chlorophyl during the phytochemical analysis and the experimental study. Therefor the following aims were: 1) Carrying out the phytochemical analysis in leaves and stems of *J. macrantha* by Liquid Chromatography-Mass Spectrometry (LC-ESI-MS/MS). 2) Determining the total phenolic compounds (TPC) and total flavonoids (TF), 3) Evaluating the antioxidant capacity using the DPPH, ABTS and FRAP methods, and 4) Evaluating the ameliorative effect of the ethyl acetate fraction from leaves and stems (LEAF and SEAF) of *J. macrantha* on ketamine-induced erectile dysfunction in rats.

## 2. Results

### 2.1. Phytochemical Analysis of the Ethyl Acetate Fraction of Leaves and Stems of J. macrantha

Phytochemical analysis was carried out by LC-ESI-MS/MS for the leaves and stems of *J. macrantha*. Our results indicated that the leaves extract had 77 phytochemical constituents, of which 25 were observed in ESI (−), 42 in ESI (+), and 10 in both modes (Table 1). In the stems extract, 42 compounds were determined, of which 18 metabolites were observed in ESI (−), 21 in ESI (+), and 3 in both modes, as presented in Table 2 and Table 3. Figure 1 and Figure 2 show the ESI-positive (+) and (−) negative chromatographic profiles for both leaves and stems of *J. macrantha.*

The retention times (Rt), adductions, experimental, and theoretical *m/z* values, ppm error, MS/MS spectrum (*m/z*: absolute intensity), SMILES (simplified molecular input line entry system) string, InChIKey (IUPAC international chemical identifier), and tentative com-pounds are available in the Appendix A. 

The phytochemical constituents determined in the extracts of leaves (Table 2) were classified as: Flavonoids (21); Coumarins and derivatives (11); Sesquiterpene lactones and sesquiterpenoids (4); terpene lactones and terpenoids (3); organic acids (3); anthraquinones (2); eudesmanolides and derivatives (2); phytoprostane (3), phenolic acids (1), others (27). In the stems (Table 3), the phytochemical constituents were classified as: coumarins and derivatives (6); flavonoids (3); sesquiterpene lactones and sesquiterpenoids (3); terpene lactones and terpenoids (3); organic acids (3); aromatic monoterpenoids (2); benzyl alcohols (2); alkyl-phenylketones (2); and others (18).

### 2.2. Total Phenolic Content, Total Flavonoids and Antioxidant Activity of J. macrantha

Regarding the antioxidant capacities of *J. macrantha* leaves and stems are observed that the leaves have a higher antioxidant capacity than stems ethyl acetate fraction and were demonstrated by the DPPH, ABTS, and FRAP assays (Table 4), these differences were statistically significant (*p* < 0.05 in Paired sample t-test) in DPPH and FRAP. Otherwise, TPC and TF in leaves were higher content than stems.

### 2.3. Effect of the Ethyl Acetate Fraction from Leaves and Stems of J. macrantha on Ketamine-Induced Erectile Dysfunction in Rats

Before the pre-copulatory actions of rats treated with *J. macrantha* and sildenafil citrate, some signs and sexual behaviors were recorded and were shown in Figure 3. Here, it is observed some behaviors such as doing circling around, body-sniffing, anogenital exploration, ear-wiggling, lordosis, hopping, mounting, and grooming. All these findings were recorded using a video camera to avoid any interference or disturbance.

In Figure 4A is observed that in mount frequency at doses of 50 mg/kg of leaves ethyl acetate fraction (LEAF-50; *p* < 0.0001) presented 11 mounts, higher than the other concentrations including all groups administrated with *J. macrantha* stems, but lower than sildenafil, which revealed an average of 13.75 mounts. SEAF-25 did not have a significant difference with the positive control group (*p* = 0.8722). Regarding Figure 2B, intromission frequency (IF) in rats administrated with LEAF-100 (*p* < 0.0001) and stems ethyl acetate fraction at 50 mg/kg (SEAF-50; *p* < 0.0001) had 11.38 and 7.88 penetrations, respectively, being more than positive control (PC) group, meanwhile sildenafil had 13.50 penetrations. Other contrary, SEAF-25 did not show a significant difference with PC group (*p* = 0.1555). On the other hand, it is observed an increase in ejaculation frequency with LEAF-25, LEAF-50, and SEAF-50 (Figure 4C) compared to PC group (*p* < 0.0001). Treatments with LEAF-50 and SEAF-50 (Figure 4D,E) showed a decrease in mount latency and intromission latency, which means increased sexual motivation and stimulation in rats, compared with sildenafil which had better effect (*p* < 0.0001).

In Figure 4F, on ejaculatory latency (EL), there was a decrease in time with all doses from leaves and stems but better at doses of 50 mg/kg (leaves and stems; *p* < 0.0001). Therefore, decreasing this parameter would improve the ejaculation in rats. The opposite happens with the increase of this indicator EL (*p* < 0.0001), which is an indicator of ejaculatory difficulty. Figure 4G, on post-ejaculatory latency is a parameter that measures the recovery time between one mating session to another, as is observed the recovery time was between 6 to 10 min in rats treated with LEAF-25 (*p* < 0.0001), LEAF-50 (*p* = 0.025), and LEAF-100 (*p* < 0.0001), respectively, and were higher than sildenafil group. Whilst SEAF-100 did not show a significant difference with PC group (*p* = 0.1682).

### 2.4. Evaluation of the Vasodilator Effect in Rats Treated with J. macrantha and Sildenafil

The evaluation of the vasodilator effect was measured as the tension expressed in grams on rat’s penile tissue. As is shown in Figure 5A, acetylcholine was not inhibited by LEAF, SEAF, and sildenafil citrate. In Figure 5B adrenaline was not inhibited by the different concentrations of SEAF and LEAF but sildenafil had better effect on adrenergic receptors (*p* = 0.0034). In Figure 5C, Calcium chloride (CaCl_2_) solution was used as an agonist of calcium channels and was only inhibited by sildenafil citrate (*p* < 0.0001). 

## 3. Discussion

For the induction of erectile dysfunction, male rats were administered intraperitoneally with ketamine at a dose of 50 mg/kg/day for a period of 14 days and exposed to a dim light (1-watt fluorescent tube) for 5 days before experiment. As can be observed, ketamine was used to produce erectile dysfunction presenting dissociative, psychotomimetic, cognitive, and peripheral side effects, which are involved in short or long-term administration as well as its dissociative use. Ketamine is an antagonist at N-methyl-D-aspartate receptors, glutamate receptors that are mainly expressed in the hippocampus and prefrontal cortex. In a study, ketamine induced the increase of stress oxidative markers such as malondialdehyde (MDA), catalase (CAT), and reduced the total antioxidant capacity (TAC) in rats, proposing that the erectile and testicular dysfunction would be through oxidative stress and decreasing the serum testosterone and luteinizing hormones [23]. Other study revealed that ketamine at 100 mg/kg/day in rats induced erectly disfunction activating apoptosis by up-regulating of inducible nitric oxide synthase (iNOS) leading to the loss of corporal smooth muscle content and reducing of neuronal nitric oxide synthase (nNOS) expression on cavernous nerve [24]. 

Of the phytochemical constituents determined in leaves and stems of *J. macrantha* ethyl acetate fraction, some of them have demonstrated to ameliorate the ED in experimental animals. Scopoletin showed a positive effect in penile erection of rats through the NO-cGMP and adenylyl cAMP-PKA signaling pathways [25]. Vitexin at doses of 40 mg/kg regulated some endocrine hormones and had an improving in the fertility rate sex performance in diabetic mice, the mechanism involved might be in the modulation of hypothalamus–pituitary–gonadal axis [26]. On the other hand, quercetin, naringenin, aromadendrin, taxifolin, and kaempferol determined in the ethanol extract from *Anaxagorea luzonensis* had an inhibitory effect on phosphodiesterase 5 [27]. In a study, moringa extract containing phenols and flavonoids such as gallic acid, catechin, chlorogenic acid, ellagic acid, quercitrin, isoquercitrina, quercetin, rutin, kaempferol, and epicatechin inhibited enzymes as arginase and angiotensin converting enzyme 2 (ACE-2), which were linked to erectile dysfunction and oxidative stress in rat’s penile [28]. 

Oxidative stress generated during inflammation, damaged tissue or chronical diseases have been associated as main key factors in the pathogenesis of ED. A previous study showed that sildenafil reduced the malondialdehyde (MDA) levels and increased the activity of antioxidant enzymes in rats with ED. Hence, some mechanisms of ED drugs are involved with the decrease of free radical levels and increase the antioxidant capacity, i.e., superoxide dismutase (SOD) and CAT [29]. In other study, paroxetine a selective serotonin reuptake inhibitor antidepressant drug significantly increased malondialdehyde (MDA) levels in rats’ penile tissues producing ED. Hereby, free radicals probably generated by antidepressant drugs would attack to the penile tissues producing lipid peroxidation and high MDA levels [30]. Thus, antioxidant compounds found in natural products might ameliorate this condition and protect to penile tissue of stress oxidative. Regarding to the mechanism stablished in our study, it can be demonstrated that SEAF and LEAF did not relax the penile tissue when were exposed to acetylcholine, however, SEAF and LEAF at 0.1 mg/mL had a stimulant effect. It is known that cholinergic stimulation of the cavernous nerve leads to increased blood flow within the penis, on the contrary the adrenergic stimulation reduces blood flow leading to the flaccid state [31], according to our results, a relaxation was observed with sildenafil whilst SEAF and LEAF did not reveal any effect on adrenergic receptors. On the other hand, the exposition to CaCl_2_ leads to extracellular calcium influx after the opening of calcium channels [32]. Although, non-significant differences were observed in LEAF and SEAF, at 0.1 mg/mL antagonized the Ca^2+^-induced contractions in rat’s penile tissue. As main limitation to understand the main mechanism involved in the vasodilator effect, Nitric oxide pathways were not evaluated. However, due to presence of several phytochemicals determined in LEAF and SEAF of *J. macrantha*, multiple mechanism could be linked to the relaxation of rat’s penile tissue and produce erection during the sexual activity. 

## 4. Materials and Methods

### 4.1. Collection of the Botanical Species

Leaves and stems of *J. macrantha* were collected in Sacrapa of the province of Paucar del Sara Sara (15°16′47″ S, 73°20′45″ W; 2691 m.a.s.l.) of the department of Ayacucho, Peru, in January 2020. The botanical identification was carried out in the Natural History Museum of the Universidad Nacional Mayor de San Marcos with Id. 039-USM-2020.

### 4.2. Preparation of the Ethanolic Extract and Ethyl Acetate Fraction

The leaves and stems were dried at room temperature for 2 weeks, the ground dry samples were macerated for 14 days with 10 L of 80 % ethanol with constant stirring. Then, filtered and concentrated using a R-300^®^ rotary evaporator (Buchi, Flawil, Switzerland) until dryness. For obtaining the phenolic fraction of leaves and stems, the hydroalcoholic extracts were suspended in distilled water, then degreased with petroleum ether (in order to eliminate fats, waxes, pigments and other metabolites that may interfere with the extraction of flavonoids). Then a liquid-liquid extraction was carried out with 300 mL of ethyl acetate using a separatory funnel that was left for a period of 24 h, then the ethyl acetate fraction was evaporated until dryness. Each ethyl acetate fraction of leaves and stems (LEAF and SEAF) were refrigerated until further use at 4 °C.

### 4.3. Phytochemical Analysis by LC-ESI-MS/MS of the Main Constituents of the Ethyl Acetate Fraction of Leaves and Stems of J. macrantha

#### 4.3.1. Preparation of the Sample

The ethyl acetate fraction of the leaves and stems of *J. macrantha* were weighed and diluted with ethyl acetate until a final concentration of 10 mg/mL had been obtained. Next, each sample was vortexed for 1 min and subsequently centrifuged for 10 min at 10,000 rpm. Finally, 800 µL of the 1 mg/mL solution supernatant (methanol:water, 1:1) was removed in vials for LC-MS analysis in a Dionex UltiMate 3000 liquid chromatograph (Thermo Fisher Scientific, San José, CA, USA) coupled to a Thermo QExactiveTM Plus Orbitrap mass spectrometer (Thermo Fisher Scientific, Bremen, Germany) with an electrospray ionization source. 

#### 4.3.2. Chromatographic Conditions

This analysis used a chromatographic column Thermo Scientific Syncronis RP-C18 (50 mm × 2.1 mm × 1.9 µm). Solvent A was 0.1% formic acid in water and 5 mM ammonium formate, and Solvent B was 0.1% formic acid in MeOH. The gradient elution of the method was as follows: 0–0.5 min, B 30%; 0.5–10.0 min, B 98%; 10.0–15.0 min, B 98%; 15.0–15.1 min, B 30%; 15.1–19.0 min, B 30%. The flow rate was 350 µL min−1 with injection of 3 µL and a column oven temperature of 40 °C.

#### 4.3.3. Mass Spectrometry Conditions

A full scan experiment combined with a fragmentation experiment (MS/MS) was per-formed for both electrospray ionization modes (ESI + and −). The ESI source parameters were as follows: spraying voltage: 3.9 kV (+) and 3.6 kV (−); envelope gas flow rate: 50 (arbitrary values); auxiliary gas flow: 10 (arbitrary values); tube lens voltage: 50 V; probe heater temperature: 400 °C; capillary temperature: 300 °C.

1. (ESI +) mode: full MS mode parameters: 35,000 resolution; ACG target (automatic gain control): 5e5; maximum IT (injection time): 100 ms; scan range: 100–1200 *m/z*. 

Dd-MS2 (data-dependent acquisition experiment, DDA) mode parameters: 17,500 resolution; ACG objective: 1e5; maximum IT: 50 ms; loop count, 3; isolation window: 1–2 *m/z*; topN, 3; NCE (stepped normalized collision energy): 15, 30, and 40.

2. (ESI −) mode: full MS mode parameters: 35,000 resolution; ACG objective: 5e5; maximum IT: 100 ms; range, 100–1200 *m/z*.

Dd-MS2 (data-dependent acquisition experiment, DDA) mode parameters: 17,500 resolution; ACG objective: 1e5; maximum IT: 50 ms; loop count, 3; isolation window: 1–2 *m/z*; topN: 3; NCE: 15, 20, and 40.

Data acquisition and processing were performed with Thermo XcaliburTM software version 3.0 (Thermo Fisher Scientific Inc., San José, CA, USA) with the Qual Browser, and metabolite annotations were performed with MS-Dial software version 4.70 (Riken, Osaka University, Ja-pan) using the MS-Dial metabolomics MPS spectral kit library (available at: http://prime.psc.riken.jp/compms/msdial/main.html; last updated on 13 April 2021).

### 4.4. Determination of Total Phenolic Compounds (TPC)

In total, 50 μL of the ethyl acetate fraction of leaves and stems (10 mg/mL) were mixed with 1 mL of distilled water, 0.5 mL of 0.2 N Folin–Ciocalteu reagent, and 2.5 mL of 5% sodium carbonate, then the sample was allowed to react in the darkness for 40 min at room temperature (20 °C). The absorbance was read at 725 nm using a Genesys 150 spectrophotometer (Thermo Scientific, Waltham, MA, USA). A standard curve was made with a gallic acid solution (50 μg/mL) at concentrations of 10, 20, 30, 40, and 50 μg/mL. The results are presented in mg gallic acid equivalent per g of extract (mg GAE/g of extract) [30].

### 4.5. Determination of Total Flavonoids

In total, 0.5 mL of ethyl acetate fraction of leaves and stems (10 mg/mL) were mixed with 1 mL with distilled water and 0.15 mL of 5% sodium nitrite; 5 min later, 0.15 mL of 10% aluminum chloride was added, then at 6 min, 2 mL of 4% sodium hydroxide was added. The sample was made up to 5 mL with distilled water, mixed, and allowed to react in the darkness for 15 min at room temperature. The absorbance was read at 510 nm against a blank using a Genesys 150 spectrophotometer (Thermo Scientific, Waltham, MA, USA). A standard curve was made with quercetin (200 μg/mL) at concentrations of 40, 80, 120, 160, and 200 μg/mL. The flavonoid content is presented as mg quercetin equivalent per g of extract (mg QE/g of extract) [31].

### 4.6. Determination of the Antioxidant Capacity by the Free Radical Sequestration Method with 2,2-Diphenyl-1-Picrylhydrazyl

For this assay, 150 μL of ethyl acetate fraction of leaves and stems (10 mg/mL) were mixed with 2850 μL of a methanolic solution of DPPH radicals (20 mg/L) with the absorbance adjusted to 0.6 ± 0.02 nm. After mixing, the sample was incubated in the dark for 30 min and the absorbance was read at 515 nm using a Genesys 150 spectrophotometer (Thermo Scientific, Waltham, MA, USA). The standard curve was elaborated with Trolox at concentrations of 0 to 800 μmol/mL [32]. The Trolox equivalent antioxidant capacity (TEAC) was expressed as μmol Trolox equivalent per gram of extract (µmol TE/g of extract).

### 4.7. Determination of the Antioxidant Capacity by the Sequestration Method with the Radical Cation of the 2.2’-Azinobis-(3-Ethylbenzothiazoline)-6-Sulfonic Acid

A standard solution (ST) was prepared by mixing 10 mL of ABTS (4.06 mg/mL) with 10 mL of potassium persulfate (0.7 mg/mL) and reacted for 12 h. The working solution (ST) was prepared with 1 mL of each extract and 60 mL of methanol. The absorbance was adjusted to 0.7 ± 0.02 with methanol at a wavelength of 734 nm, then 150 μL of the ethyl acetate fraction of leaves and stems (5 mg/mL) were mixed with 2850 μL of the working solution and incubated in the dark for 7 min, followed by reading the absorbances at 734 nm using a Genesys 150 spectrophotometer (Thermo Scientific, Waltham, MA, USA) [33]. The standard curve was made with Trolox at 0–400 μmol/mL. The Trolox equivalent antioxidant capacity (TEAC) was expressed as μmol Trolox equivalent per gram of extract (µmol TE/g of extract).

### 4.8. Determination of the Antioxidant Capacity by the Ferric Reducing Antioxidant Power (FRAP) Method

For the determination, 150 μL of sample was mixed with 2850 μL of the previously prepared 2,3,5-triphenyltetrazolium chloride (TPTZ) reagent, which was left for 30 min at 37 °C. The absorbance was measured at 593 nm using a Genesys 150 spectrophotometer (Thermo Scientific, Waltham, MA, USA). The standard curve was made with Trolox from 50 to 800 μmol/mL. The results were expressed as μmol Trolox equivalent per gram of extract (μmol TE/g of extract) [33].

### 4.9. Effect of the Ethyl Acetate Fraction from Leaves and Stems of J. Macrantha on Ketamine-Induced Erectile Dysfunction in Rats

#### Evaluation of Erectly Dysfunction

The animals were prepared and conditioned by the method of Shang et al., [24]. In this study, 100 females and males Holtzman rats weighing 200 ± 50 g were acquired from the Bioterium of the Universidad Nacional Agraria La Molina (UNALM) and transported to the city of Ayacucho and adapted in the Bioterium of the School of Pharmacy and Biochemistry of the Universidad Nacional de San Cristóbal de Huamanga (UNSCH). Then, nine groups of eight animals each were grouped as follows:(a)Group I: distilled water at doses of 10 mL/kg was administered orally, which served as negative control (NC).(b)Group II: Ketamine at doses of 50 mg/kg was administered intraperitoneally, which served as a positive control (PC).(c)Group III: Ketamine at 50 mg/kg (IP) plus leaves ethyl acetate fraction at doses of 25 mg/kg (LEAF-25) by oral administration.(d)Group IV: Ketamine at 50 mg/kg (IP) plus leaves ethyl acetate fraction at doses of 50 mg/kg (LEAF-50) by oral administration.(e)Group V: Ketamine at 50 mg/kg (IP) plus leaves ethyl acetate fraction at doses of 100 mg/kg (LEAF-100) by oral administration.(f)Group VI: Ketamine at 50 mg/kg (IP) plus stems ethyl acetate fraction at doses of 25 mg/kg (SEAF-25) by oral administration.(g)Group VII: Ketamine at 50 mg/kg (IP) plus stems ethyl acetate fraction at doses of 50 mg/kg (SEAF-50) by oral administration.(h)Group VIII: Ketamine at 50 mg/kg (IP) plus stems ethyl acetate fraction at doses of 100 mg/kg (SEAF-100) by oral administration.(i)Group XI: Ketamine 50 mg/kg (IP) plus sildenafil citrate at 5 mg/kg by oral administration.

For the induction of erectile dysfunction, male rats were administered intraperitoneally with ketamine at doses of 50 mg/kg/day for a period of 14 days and were exposed to a dim light (1-watt fluorescent tube) for 5 days prior to experiment. After induction, on day 20, experimental groups were administrated with *J. macrantha* and sildenafil citrate by oral administration in a single dose, a screening of sexual behavior was performed by evaluating the frequencies of mounting, intrusion and ejaculation, such as the respective latencies of mounting, intrusion, ejaculation and post-ejaculation. For evaluating sexual behavior, a time interval of 15 min per rat was used, in which the respective frequencies and latencies were evaluated. Female rats were administered subcutaneously 48 h and 4 h with estradiol benzoate at 10 µg/100 g of body weight and progesterone at doses of 0.5 mg/100 g of body weight, before putting them together with male rats. Then, a receptive female was introduced to the male cages. Observation of mating was carried out at 8:00 p.m.

The parameters to measure male sexual behavior were:(a)Mount frequency (MF): It is the number of mounts without penetration at the moment that female rat is introduced to the male’s cage until ejaculation.(b)Intromission frequency (IF): It is the number of penetrations at the moment that female rat is introduced to the male’s cage until ejaculation.(c)Ejaculation frequency (EF): It is the time interval between the introduction of the female rat and the first mount of the male rat.(d)Mount latency (ML): It is the time interval that elapses when the female rat is introduced to the cage, until the male performs the first mount of the copulatory series.(e)Intromission latency (IL): It is the time interval at the moment of introduction of the female rat to the first penetration of male rat. It is usually characterized by pelvic thrust and jump to dismount.(f)Ejaculatory latency (EL): It is the time interval between the first penetration and ejaculation. It is usually characterized by a prolonged, deep pelvic thrust and slow dismounting followed by a period of inactivity or reduced activity.(g)Post ejaculatory latency (PEL): It is the time interval between ejaculation and the first penetration of the following series.

The Local Committee of Ethics on Animal Experimentation approved all experimental procedures, with i.d. 261-2020-FCSA-UNSCH.

### 4.10. Evaluation of the Vasodilator Effect

For the determination of the vasodilator effect, the corpus cavernosum smooth muscle was kept in vital condition, providing the necessary elements for its maintenance and stability. The corpus cavernosum was isolated and poured into the Krebs-Henseleit (KHS) nutrient liquid with oxygenation 95.0% O_2_ and 5.0% CO_2_ for 30 min. Two strips of the smooth muscle of the corpus cavernosum (10 to 12 mm in length and 1–2 mm in thickness) were isolated by dissecting part of the penis, working with a solution containing KHS (pH 7.3) with the following composition: NaCl = 7.01 g/L, KCl = 0.34 g/L, KH_2_PO_4_ = 0.1 g/L, NaHCO_3_ = 1.99 g/L, CaCl_2_ = 0.2 g/L, MgSO_4_ = 0.3 g/L and glucose 1.8 g/L, eight repetitions were performed with the ethyl acetate fractions of *Jatropha macrantha* leaves and stems at concentrations of 0.1; 0.5 and 1.0 mg/mL and sildenafil 3.2 × 10^−5^ mg/mL as a positive control. To achieve maximum contractions, acetylcholine, adrenaline and calcium chloride (CaCl_2_) were administered at 5 × 10^−2^ M, 1.0 µg/mL, and 17.8 mg/mL, respectively. The data were expressed in tension (g) by reading contraction or relaxation.

### 4.11. Data Analysis

The results are presented as the means plus standard deviation of eight animals per group. The differences between the means were analyzed using ANOVA followed by Tukey and Dunnett’s Test for erectile dysfunction study and for TPC, TF, and antioxidant capacity, were used paired sample t-test using Graph Pad Prism v6 software. A *p*-value less than 0.05 is considered significant.

## 5. Conclusions

Based on our results, the ethyl acetate fraction of *Jatropha macrantha* leaves at 50 mg/kg/day by oral administration presented an ameliorative effect on ketamine-induced erectile dysfunction in rats. Furthermore, it showed a high content of total phenols and flavonoids than stems ethyl acetate fraction, which leads to a high antioxidant capacity in DPPH, ABTS and FRAP assays. In the analysis carried out with LC-ESI-MS/MS, leaves and stems contained 77 and 42 phytochemical constituents, respectively. Some chemical groups highlighted were organic acids, phenolic acids, flavonoids, coumarins, fatty acids, lipids, sesquiterpene lactones, terpenoids and anthraquinones. *J. macrantha* might be useful as a promising herbal medicine in erectile dysfunction or aphrodisiac.

## Figures and Tables

**Figure 1 molecules-27-00115-f001:**
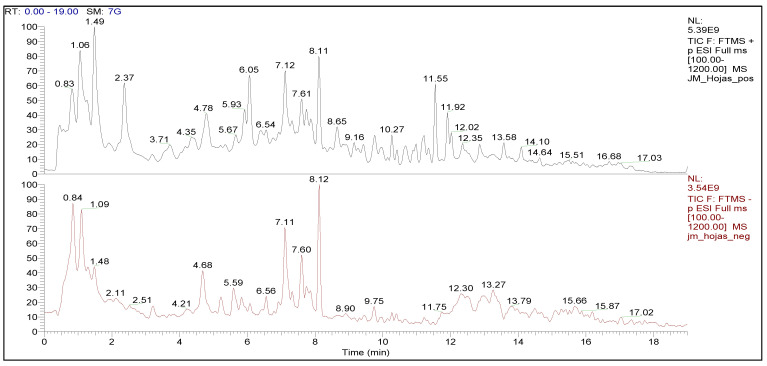
Chromatographic profile (LC-ESI-MS/MS) of leaves of *J. macrantha* ethyl acetate fraction, in both ESI (+) and ESI (−) ionization modes.

**Figure 2 molecules-27-00115-f002:**
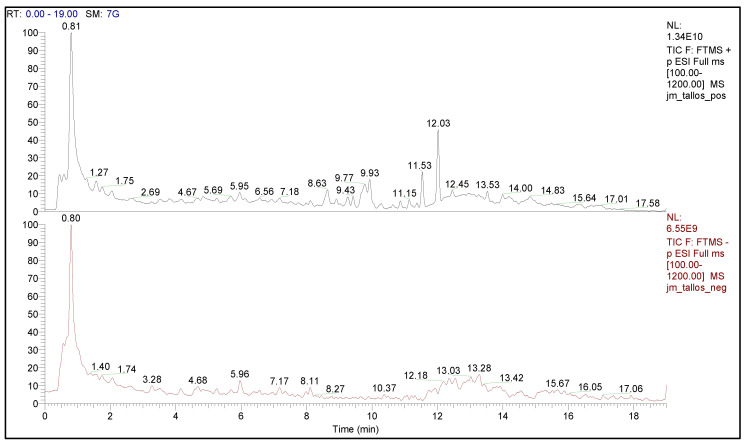
Chromatographic profile (LC-ESI-MS/MS) of stems of *J. macrantha* ethyl acetate fraction, in both ESI (+) and ESI (−) ionization modes.

**Figure 3 molecules-27-00115-f003:**
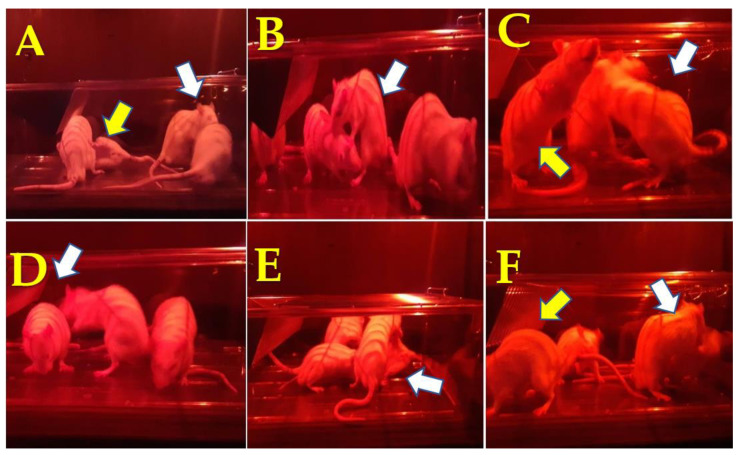
Distinctly visible signs of pre-copulatory actions of rats treated with *J. macrantha* and sildenafil. (**A**): circling around (yellow arrow), and body-sniffing (white arrow), (**B**): mounting (white arrow), (**C**): ear-wiggling (white arrow) and grooming (yellow arrow), (**D**): anogenital exploration (white arrow), (**E**): circling around (white arrow), (**F**): lordosis (yellow arrow), and mounting (white arrow).

**Figure 4 molecules-27-00115-f004:**
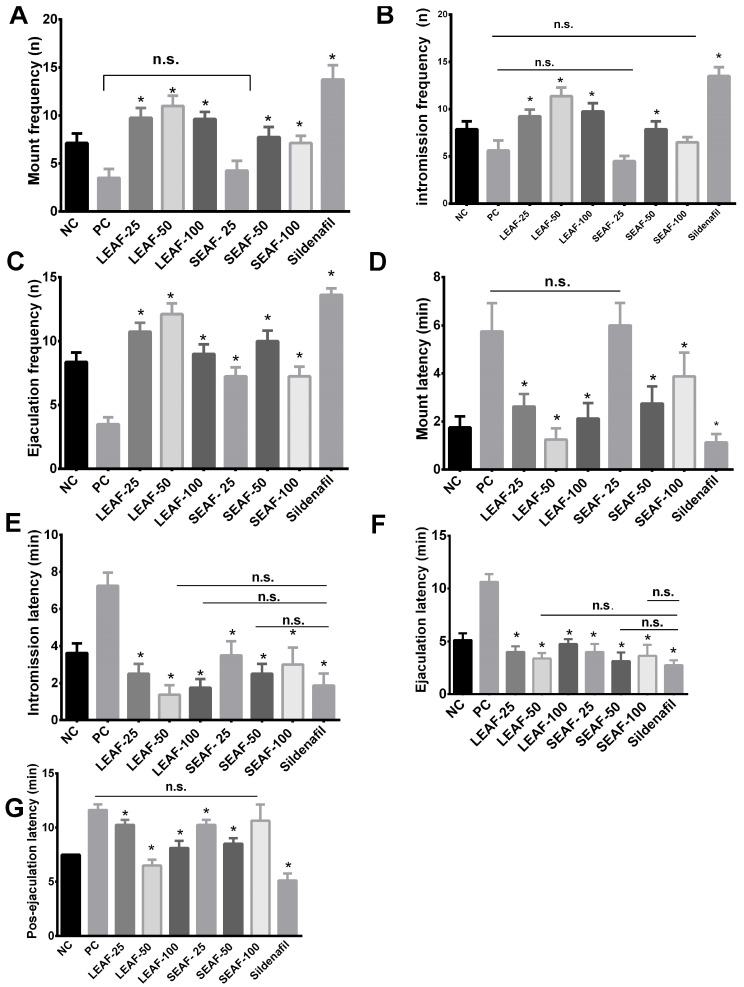
Effect of the ethyl acetate fraction from leaves and stems of *J. macrantha* Müll Arg. on ketamine-induced erectile dysfunction in rats. (**A**). Mount frequency, (**B**). Intromission frequency, (**C**). Ejaculation frequency, (**D**). Mount latency, (**E**): Intromission latency, (**F**). Ejaculation latency, (**G**). Post-ejaculation latency. NC: negative control, PC: positive control; LEAF-25: Leaves ethyl acetate fraction 25 mg/kg; LEAF-50: Leaves ethyl acetate fraction 50 mg/kg; LEAF-100: Leaves ethyl acetate fraction 100 mg/kg; SEAF-25: stems ethyl acetate fraction 25 mg/kg; SEAF-50: stems ethyl acetate fraction 50 mg/kg; SEAF-100: stems ethyl acetate fraction 100 mg/kg; * *p* < 0.05 Tukey Test; n.s.: non-significant difference.

**Figure 5 molecules-27-00115-f005:**
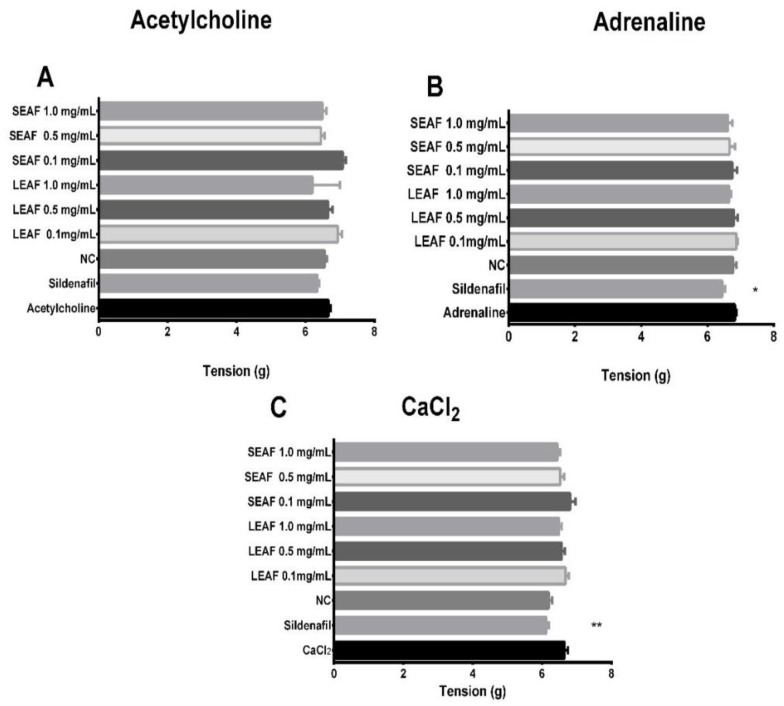
Vasodilator effect in rats treated with *J. macrantha* and sildenafil. NC: negative control, PC: positive control; LEAF-25: Leaves ethyl acetate fraction 25 mg/kg; LEAF-50: Leaves ethyl acetate fraction 50 mg/kg; LEAF-100: Leaves ethyl acetate fraction 100 mg/kg; SEAF-25: stems ethyl acetate fraction 25 mg/kg; SEAF-50: stems ethyl acetate fraction 50 mg/kg; SEAF-100: stems ethyl acetate fraction 100 mg/kg; * *p* < 0.05 (Dunnet’s Test) compared to adrenaline; ** *p* < 0.001 (Dunnet’s Test) compared to CaCl_2_; n.s.: non-significant difference.

**Table 1 molecules-27-00115-t001:** Number of annotated metabolites (via MS and MS/MS) in each extract according to the ESI (−) and ESI (+) ionization modes.

Ethyl Acetate Fraction	ESI (−)	ESI (+)	ESI (+/−)	Total
Leaves	25	42	10	77
Stems	18	21	3	42

**Table 2 molecules-27-00115-t002:** Phytochemical constituents of leaves of *J. macrantha* ethyl acetate fraction determined by LC-ESI-MS/MS.

N°	Retention Time (Min)	Theoretical Mass (Neutral Form)	Molecular Formula (Neutral Form)	Predicted Metabolite	Chemical Group
1	0.41	104.10754	C_5_H_14_NO	Choline	Cholines
2	0.41	117.07898	C_5_H_11_NO_2_	Betaine	Alpha amino acids
3	0.45	122.04801	C_6_H_6_N_2_O	Niacinamide	Vitamin B₃
4	0.46	118.02661	C_4_H_6_O_4_	Succinic acid	Dicarboxylic acids and derivatives
5	0.54	99.06841	C_5_H_9_NO	2-Piperidone	Piperidinones
6	0.55	110.03678	C_6_H_6_O_2_	Catechol	Catechols
7	0.66	184.03717	C_8_H_8_O_5_	Methylgallate	Galloyl esters
8	0.69	138.03169	C_7_H_6_O_3_	Salicylic acid	Salicylic acids
9	0.70	178.02661	C_9_H_6_O_4_	Esculetin Syn. 6,7-Dihydroxycoumarin	Dihydroxycoumarins
	0.71	178.02661	C_9_H_6_O_4_		
10	0.73	152.04734	C_8_H_8_O_3_	2,4-Dihydroxyacetophenone	Alkyl-phenylketones
11	0.74	121.05276	C_7_H_7_NO	Benzamide	Benzamides
12	0.78	564.14791	C_26_H_28_O_14_	NP-000004 Syn. Apigenin 6-C-glucoside 8-C-arabinoside	Flavonoid C-glycosides
	0.79	564.14791	C_26_H_28_O_14_		
13	0.79	208.03717	C_10_H_8_O_5_	Fraxetin	Dihydroxycoumarins
14	0.84	448.10056	C_21_H_20_O_11_	Homoorientin Syn. Luteolin-6-C-glucoside	Flavonoid C-glycosides
15	0.84	448.10056	C_21_H_20_O_11_	Luteolin-8-C-glucoside Syn. Orientin	Flavonoid C-glycosides
	0.85	448.10056	C_21_H_20_O_11_		
16	0.89	122.03678	C_7_H_6_O_2_	3-Hydroxybenzaldehyde	Phenolic acids
17	0.90	147.06841	C_9_H_9_NO	Indole-3-carbinol	3-alkylindoles
18	0.90	174.07931	C_10_H_10_N_2_O	Indole-3-acetamide	3-alkylindoles
19	0.93	282.14672	C_15_H_22_O_5_	5,9-dihydroxy-7-(hydroxymethyl)-5,7-dimethyl-4,5a,6,8,8a,9-hexahydro-1H-azuleno[5,6-c]furan-3-onenSyn. Lactarorufin B	Lactarane sesquiterpenes
20	1.04	192.04226	C_10_H_8_O_4_	Scopoletin	Hydroxycoumarins
	1.05	192.04226	C_10_H_8_O_4_		
21	1.07	222.05282	C_11_H_10_O_5_	Isofraxidin	Hydroxycoumarins
22	1.08	222.05282	C_11_H_10_O_5_	8-hydroxy-6,7-dimethoxy-2H-chromen-2-oneSyn. Fraxidin	Hydroxycoumarins
23	1.10	432.10565	C_21_H_20_O_10_	Vitexin n (Isomer I)	Flavonoid C-glycosides
	1.11	432.10565	C_21_H_20_O_10_		
24	1.17	196.10994	C_11_H_16_O_3_	Loliolide	Benzofurans
25	1.32	432.10565	C_21_H_20_O_10_	Vitexin Syn. Flavone, 8-D-glucosyl-4’,5,7-trihydroxy-	Flavonoid C-glycosides
26	1.31	432.10565	C_21_H_20_O_10_	Isovitexin Syn. Homovitexin	Flavonoid C-glycosides
	1.32	432.10565	C_21_H_20_O_10_		
27	1.40	448.10056	C_21_H_20_O_11_	Kaempferol-7-O-glucoside	Flavonoid O-glycosides
28	1.41	448.10056	C_21_H_20_O_11_	NCGC00385820-01!5,7-dihydroxy-2-[3-hydroxy-4-[(2S,3R,4S,5S,6R)-3,4,5-trihydroxy-6-(hydroxymethyl)oxan-2-yl]oxyphenyl]chromen-4-one Syn. Luteolin 4’-O-glucoside (Isomer I)	Flavonoid O-glycosides
29	1.48	145.05276	C_9_H_7_NO	2-hydroxyquinoline	Hydroquinolones
30	1.49	145.05276	C_9_H_7_NO	Indole-3-carboxyaldehyde	Indoles
31	1.49	196.10994	C_11_H_16_O_3_	Loliolide	Benzofurans
32	1.59	206.05791	C_11_H_10_O_4_	(4S,5Z,6S)-4-(2-methoxy-2-oxoethyl)-5-[2-[(E)-3-phenylprop-2-enoyl]oxyethylidene]-6-[(2S,3R,4S,5S,6R)-3,4,5-trihydroxy-6-(hydroxymethyl)oxan-2-yl]oxy-4H-pyran-3-carboxylic acid Syn. Jasminoside	Coumarins and derivatives
33	1.60	288.06339	C_15_H_12_O_6_	(2S,3S)-3,5,7-trihydroxy-2-(4-hydroxyphenyl)-2,3-dihydrochromen-4-one Syn. (-)-dihydrokaempferol	Flavonoids
34	1.71	175.06333	C_10_H_9_NO_2_	3-Indoleacetic acid	Indole-3-acetic acid derivatives
35	1.91	164.04734	C_9_H_8_O_3_	Coumaric acid (Isomer I)	Coumaric acid and derivatives
36	2.00	416.11073	C_21_H_20_O_9_	Puerarin Syn. Daidzein-8-C-glucoside	Isoflavonoid C-glycosides
	2.01	416.11073	C_21_H_20_O_9_		
37	2.26	432.10565	C_21_H_20_O_10_	Apigetrin Syn. Apigenin 7-O-glucoside	Flavonoid O-glycosides
38	2.27	432.10565	C_21_H_20_O_10_	Aloenin	Anthraquinones
39	2.31	164.04734	C_9_H_8_O_3_	Coumaric acid (Isomer II)	Coumaric acid and derivatives
40	2.31	178.06299	C_10_H_10_O_3_	Coniferylaldehyde	Methoxyphenols
41	2.54	448.10056	C_21_H_20_O_11_	Luteolin-7-glucoside (Isomer I)	Flavonoid O-glycosides
42	2.54	448.10056	C_21_H_20_O_11_	NCGC00385820-01!5,7-dihydroxy-2-[3-hydroxy-4-[(2S,3R,4S,5S,6R)-3,4,5-trihydroxy-6-(hydroxymethyl)oxan-2-yl]oxyphenyl]chromen-4-oneSyn. Luteolin 4’-O-glucoside (Isomer II)	Flavonoid O-glycosides
43	3.11	332.18350	C_16_H_28_O_7_	2-(hydroxymethyl)-6-(6-hydroxy-6-methyl-3-propan-2-ylcyclohex-3-en-1-yl)oxyoxane-3,4,5-triol Syn. MCULE-9958171223	Fatty acyl glycosides of mono- and disaccharides
44	3.21	188.10486	C_9_H_16_O_4_	Azelaic acid	Organic acids
45	3.56	448.10056	C_21_H_20_O_11_	Luteolin-7-glucoside	Flavonoid O-glycosides
	3.57	448.10056	C_21_H_20_O_11_		
46	3.96	164.04734	C_9_H_8_O_3_	Coumaric acid (Isomer III)	Coumaric acid and derivatives
47	4.02	264.13616	C_15_H_20_O_4_	Abscisic acid	Abscisic acids and derivatives
48	4.40	432.10565	C_21_H_20_O_10_	Vitexin (Isomer II)	Flavonoid C-glycosides
49	4.62	272.06847	C_15_H_12_O_5_	Naringenin	Flavanone
50	4.68	286.04774	C_15_H_10_O_6_	Luteolin (Isomer I)	Flavones
	4.69	286.04774	C_15_H_10_O_6_		
51	5.06	262.12051	C_15_H_18_O_4_	Dihydro-8-deoxy-lactucin	Gamma butyrolactones
52	5.21	164.04734	C_9_H_8_O_3_	Coumaric acid (Isomer VI)	Coumaric acid and derivatives
53	5.23	210.12559	C_12_H_18_O_3_	Jasmonic Acid	Jasmonic acids
54	5.32	286.04774	C_15_H_10_O_6_	Luteolin(Isomer II)	Flavones
55	5.42	594.13734	C_30_H_26_O_13_	Kaempferol-3-O-glucoside-2’’-p-coumaroyl	Flavonoid O-glycosides
56	5.60	164.04734	C_9_H_8_O_3_	Coumaric acid (Isomer V)	Coumaric acid and derivatives
	5.60	164.04734	C_9_H_8_O_3_		
57	5.70	270.05282	C_15_H_10_O_5_	Apigenin	Flavones
58	5.70	270.05282	C_15_H_10_O_5_	Aloe-emodin	Anthraquinones
59	5.83	300.06339	C_16_H_12_O_6_	Diosmetin	Flavones
	5.84	300.06339	C_16_H_12_O_6_		
60	6.03	252.17254	C_15_H_24_O_3_	NCGC00169905-02_C_15_H_24_O_3__2-Pentenoic acid, 5-(3-hydroxy-2,3-dimethylbicyclo[2.2.1]hept-2-yl)-2-methyl-, (2E)-	Sesquiterpenoids
61	6.10	334.17802	C_19_H_26_O_5_	Arnicolide C	Sesquiterpene lactones
62	6.25	232.14633	C_15_H_20_O_2_	Alantolactone	Eudesmanolides, secoeudesmanolides, and derivatives
63	6.63	282.14672	C_15_H_22_O_5_	Artemisinin	Terpene lactones
64	7.12	292.20384	C_18_H_28_O_3_	NCGC00386020-01_C18H28O3_8-{(1S,5R)-4-Oxo-5-[(2Z)-2-penten-1-yl]-2-cyclopenten-1-yl}octanoic acidSyn. Chromomoric acid B (Isomer I)	Phytoprostane
65	8.28	316.20384	C_20_H_28_O_3_	Cafestol	Naphthofurans
66	8.52	250.15689	C_15_H_22_O_3_	NCGC00169029-02_C_15_H_22_O_3__Naphtho[2,3-b]furan-2(4H)-one, 4a,5,6,7,8,8a,9,9a-octahydro-9a-hydroxy-3,4a,5-trimethyl-, (4aR,5S,8aS,9aR)-	Terpene lactones
67	8.54	250.15689	C_15_H_22_O_3_	2-[(2S,4aR,8aS)-2-hydroxy-4a-methyl-8-methylidene-3,4,5,6,7,8a-hexahydro-1H-naphthalen-2-yl]prop-2-enoic acid	Eudesmane, isoeudesmane or cycloeudesmane sesquiterpenoids
68	8.76	470.33961	C_30_H_46_O_4_	NCGC00169801-02_C_30_H_46_O_4__Lanosta-8,24-dien-26-oic acid, 21-hydroxy-3-oxo-, (5xi,13alpha,14beta,17alpha,20S,24E)- Syn. MEGxp0_001112	Triterpenoids
69	8.98	292.20384	C_18_H_28_O_3_	NCGC00386020-01_C_18_H_28_O3_8-{(1S,5R)-4-Oxo-5-[(2Z)-2-penten-1-yl]-2-cyclopenten-1-yl}octanoic acidSyn. Chromomoric acid B (Isomer II)	Phytoprostane
70	9.16	292.20384	C_18_H_28_O_3_	NCGC00386020-01_C_18_H_28_O_3__8-{(1S,5R)-4-Oxo-5-[(2Z)-2-penten-1-yl]-2-cyclopenten-1-yl} octanoic acid Syn. Chromomoric acid B (Isomer III)	Phytoprostane
71	9.53	472.35526	C_30_H_48_O_4_	NCGC00385237-01_C30H48O4	Cucurbitacins
72	10.42	676.36701	C_33_H_56_O_14_	DGMG 18:3	Lipids
73	10.65	514.31418	C_27_H_46_O_9_	MGMG 18:3	Lipids
74	10.66	514.31418	C_27_H_46_O_9_	NCGC00380867-01_C_27_H_46_O_9__9,12,15-Octadecatrienoic acid, 3-(hexopyranosyloxy)-2-hydroxypropyl ester, (9Z,12Z,15Z)-	Glycosylmonoacylglycerols
	10.66	514.31418	C_27_H_46_O_9_		
75	11.18	304.24023	C_20_H_32_O_2_	NCGC00384643-01_C_20_H_32_O_2__(2E)-3-Methyl-5-[(1S,8aS)-5,5,8a-trimethyl-2-methylenedecahydro-1-naphthalenyl] -2-pentenoic acid	Diterpenoids
76	11.69	356.29266	C_21_H_40_O_4_	Monoolein	1-Monoglyceride
77	11.92	592.26857	C_35_H_36_N_4_O_5_	Pheophorbide A	Chlorins

**Table 3 molecules-27-00115-t003:** Phytochemical constituents of stems of *J. macrantha* ethyl acetate fraction determined by LC-ESI-MS/MS.

N°	Retention time (min)	Theoretical mass (neutral form)	Molecular Formula (neutral form)	Predicted metabolite	Chemical group
1	0.41	117.07898	C_5_H_11_NO_2_	Betaine	Alpha amino acids
2	0.42	135.05450	C_5_H_5_N_5_	Adenine	6-Aminopurines
3	0.42	169.07389	C_8_H_11_NO_3_	Pyridoxine	Vitamin B6
4	0.45	118.02661	C_4_H_6_O_4_	Succinic acid	Dicarboxylic acids and derivatives
5	0.46	122.04801	C_6_H_6_N_2_O	Niacinamide	Nicotinamides
6	0.51	104.04734	C_4_H_8_O_3_	2-Hydroxybutyric acid	Alpha hydroxy acids and derivatives
7	0.55	99.06841	C_5_H_9_NO	2-Piperidone	Piperidinones
8	0.56	110.03678	C_6_H_6_O_2_	Catechol	Catechols
9	0.56	132.04226	C_5_H_8_O_4_	Glutaric acid	Dicarboxylic acids and derivatives
10	0.67	138.03169	C_7_H_6_O_3_	Salicylic acid	Salicylic acids
11	0.67	145.05276	C_9_H_7_NO	2-hydroxyquinoline	Hydroquinolones
12	0.70	178.02661	C_9_H_6_O_4_	6,7-Dihydroxycoumarin Syn. Esculetin	Dihydroxycoumarins
	0.71	178.02661	C_9_H_6_O_4_		
13	0.70	196.07356	C_10_H_12_O_4_	1-(2-hydroxy-4,6-dimethoxyphenyl)ethanone Syn. Xanthoxylin	Alkyl-phenylketones
14	0.74	152.04734	C_8_H_8_O_3_	2,4-Dihydroxyacetophenone	Alkyl-phenylketones
15	0.81	208.03717	C_10_H_8_O_5_	NCGC00017270-07!7,8-dihydroxy-6-methoxychromen-2-oneSyn. Fraxetin	Dihydroxycoumarins
	0.81	208.03717	C_10_H_8_O_5_		
16	0.91	122.03678	C_7_H_6_O_2_	3-Hydroxybenzaldehyde	Phenolic acids
17	0.93	282.14672	C_15_H_22_O_5_	5,9-Dihydroxy-7-(hydroxymethyl)-5,7-dimethyl-4,5a,6,8,8a,9-hexahydro-1H-azuleno[5,6-c]furan-3-one Syn. Lactarorufin B	Lactarane sesquiterpenes
18	1.05	192.04226	C_10_H_8_O_4_	Scopoletin	Hydroxycoumarins
	1.06	192.04226	C_10_H_8_O_4_		
19	1.08	178.06299	C_10_H_10_O_3_	2-Methoxycinnamic acid	Cinnamic acids
20	1.09	222.05282	C_11_H_10_O_5_	8-Hydroxy-6,7-dimethoxy-2H-chromen-2-one Syn. Fraxidin	Hydroxycoumarins
21	1.15	432.10565	C_21_H_20_O_10_	Vitexin	Flavonoid C-glycosides
22	1.18	196.10994	C_11_H_16_O_3_	Loliolide	Benzofurans
23	1.22	132.07864	C_6_H_12_O_3_	2-Hydroxy-4-methylpentanoic acid	Hydroxy fatty acids
24	1.51	145.05276	C_9_H_7_NO	Indole-3-carboxyaldehyde	Indoles
25	1.94	150.10447	C_10_H_14_O	Cuminyl alcohol (Isomer I)	Benzyl alcohols
26	2.59	262.14164	C_12_H_22_O_6_	9-(2,3-dihydroxypropoxy)-9-oxononanoic acid	Organic acids
27	2.65	372.21480	C_19_H_32_O_7_	NCGC00384741-01_C_19_H_32_O_7__2-Cyclohexen-1-one, 3-[3-(beta-D-glucopyranosyloxy)butyl]-2,4,4-trimethyl-Syn. Megastigm-5-En-4-One 9-Glucoside	Fatty acyl glycosides of mono- and disaccharides
28	2.91	316.20384	C_20_H_28_O_3_	Cafestol	Naphthofurans
29	3.25	150.10447	C_10_H_14_O	Cuminyl alcohol (Isomer II)	Benzyl alcohols
30	3.28	188.10486	C_9_H_16_O_4_	Azelaic acid	Organic acids
31	4.63	150.10447	C_10_H_14_O	Thymol (Isomer I)	Aromatic monoterpenoids
32	4.73	286.04774	C_15_H_10_O_6_	Kaempferol / 3’,4’,5,7-tetrahydroxyflavone	Flavonols
33	5.09	262.12051	C_15_H_18_O_4_	Dihydro-8-deoxy-lactucin	Gamma butyrolactones
34	5.88	300.06339	C_16_H_12_O_6_	Diosmetin / tectorigenin	Flavonoids
35	5.71	150.10447	C_10_H_14_O	Thymol(Isomer II)	Aromatic monoterpenoids
36	7.99	348.19367	C_20_H_28_O_5_	Ingenol	Tigliane and ingenane diterpenoids
37	8.30	392.21989	C_22_H_32_O_6_	NCGC00180384-03_C_22_H_32_O_6__(1S,2R,4aR,8aR)-1-Acetoxy-7-isopropylidene-1,4a-dimethyl-6-oxodecahydro-2-naphthalenyl 2,3-dimethyl-2-oxiranecarboxylate (Isomer I)	Sesquiterpenoids
38	8.80	229.24056	C_14_H_31_NO	N,N-Dimethyldodecylamine N-oxide	Long-chain alkyl amine oxides
39	8.97	392.21989	C_22_H_32_O_6_	NCGC00180384-03_C_22_H_32_O_6__(1S,2R,4aR,8aR)-1-Acetoxy-7-isopropylidene-1,4a-dimethyl-6-oxodecahydro-2-naphthalenyl 2,3-dimethyl-2-oxiranecarboxylate (Isomer II)	Sesquiterpenoids
40	9.88	148.05243	C_9_H_8_O_2_	3,4-Dihydrocoumarin	Dihydrocoumarins
41	12.03	162.03169	C_9_H_6_O_3_	Umbelliferone	Hydroxycoumarins
42	12.05	358.30831	C_21_H_42_O_4_	1-Monostearin	1-monoacylglycerols

**Table 4 molecules-27-00115-t004:** Total phenolic compounds, total flavonoids and antioxidant activity of leaves and stems of *J. macrantha* ethyl acetate fraction.

Samples	Total Phenolic Compounds (mg GAE/g extract)	Total Flavonoids (mg QE/g)	DPPH (µmol TE/g)	ABTS (µmol TE/ g)	FRAP (µmol TE/g)
Leaves	359 ± 5.21	101 ± 1.42	796 ± 3.15	679 ± 0.85	806 ± 3.42
Stems	306 ± 1.93 *	23.7 ± 0.80 *	647 ± 3.27 *	668 ± 2.30	575 ± 2.86 *

* *p* < 0.05; Paired sample t-test between mean values of leaves and stems ethyl acetate fractions. mg GAE/g extract: milligrams of gallic acid equivalents per gram of extract. mg QE/g: milligrams of quercetin equivalents per gram of extract. µmol TE/g: micromol of trolox equivalents per gram of extract.

## Data Availability

The data that support the results and findings of this study is available from the corresponding author upon request.

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
