# Peer review of "Phytochemical Screening by LC-ESI-MS/MS and Effect of the Ethyl Acetate Fraction from Leaves and Stems of Jatropha macrantha Müll Arg. on Ketamine-Induced Erectile Dysfunction in Rats"

_molecules, 2021, doi:10.3390/molecules27010115_

Round 1

Reviewer 1 Report

The authors characterize the roots and stem of Jatropha macrantha, an endemic plant of South America with different therapeutic virtues.

The authors characterized the two extracts (leaves and stems) from a phytochemical point of view, separating and identifying different components. In addition, they evaluated the antioxidant action of the two extracts.

Finally, the authors evaluated applications in the field of erectile dysfunction in a rat model. In particular, after induction of erectile dysfunction with ketamine, rats restore their copulatory functions after treatment with J. macrantha derivatives. The plant extracts act like the control, sidenafil.

The experimental design and the statistical approach are correct.

However, some improvements should be made:

Authors must correct errors, and incomprehensible sentences in the text.

The caption of Figure 1 should be supplemented in more detail. The behaviors indicated are not clear.

The authors should explain why they treat female rats with steroid hormones. Isn't it enough to evaluate their state of estrus?

Author Response

Reviewer 1:

The authors characterize the roots and stem of Jatropha macrantha, an endemic plant of South America with different therapeutic virtues.

The authors characterized the two extracts (leaves and stems) from a phytochemical point of view, separating and identifying different components. In addition, they evaluated the antioxidant action of the two extracts.

Finally, the authors evaluated applications in the field of erectile dysfunction in a rat model. In particular, after induction of erectile dysfunction with ketamine, rats restore their copulatory functions after treatment with J. macrantha derivatives. The plant extracts act like the control, sildenafil.

The experimental design and the statistical approach are correct.

However, some improvements should be made:

Authors must correct errors, and incomprehensible sentences in the text.

R1: thank you for your comments, we reviewed the manuscript and corrected substantial errors in grammar.

The caption of Figure 1 should be supplemented in more detail. The behaviors indicated are not clear.

R2: thank you for your comments, we amended the figure which indicates the sexual behavior.

The authors should explain why they treat female rats with steroid hormones. Isn't it enough to evaluate their state of estrus?

R3: thank you for your comments, in effect,  it is enough to evaluate their estrus state, but  it supposes waiting a few time if female rats are not in its ovulation period or sexually receptive to begin our experimental study. Therefor, we accelerated that process with steroid hormones.

Reviewer 2 Report

This paper looks sound, and is of interest to the journal’s readership. I think it is acceptable after some modifications. The article is well designed mainly from an analytical point of view. I only have one question relatively to this point.The authors use ethyl acetate for liquid-liquid extraction. Did you try to use other solvents for the extraction, e.g. hexane?

I think that the authors should highlight the novelty of the proposed manuscript so that it is competitive with other publications.

Author Response

Reviewer 2:

This paper looks sound and is of interest to the journal’s readership. I think it is acceptable after some modifications. The article is well designed mainly from an analytical point of view. I only have one question relatively to this point. The authors use ethyl acetate for liquid-liquid extraction. Did you try to use other solvents for the extraction, e.g. hexane?

R1: Thank you for your comments, we used ethyl acetate because we aimed to obtain a rich extract in phenolic compounds, whilst n-hexane is used to obtain mainly non-polar compounds and would limit our metabolite extraction.

I think that the authors should highlight the novelty of the proposed manuscript so that it is competitive with other publications.

R2: Thank you for your valuable comments.

Reviewer 3 Report

The presented work „Phytochemical screening by LC-ESI-MS/MS and Effect of the Ethyl Acetate Fraction from Leaves and Stems of Jatropha macrantha Mull Arg. On Ketamine-Induced Erectile Dysfunction in Rats” is well composed. The research topic is interesting. I nave only one comment: In chapter 4. Materials and Methods - chapter 4.3 and chapter 4.9 are the same. This needs to be changed.

Therefore, I suggest minor revision.

Author Response

Reviewer 3

The presented work „Phytochemical screening by LC-ESI-MS/MS and Effect of the Ethyl Acetate Fraction from Leaves and Stems of Jatropha macrantha Mull Arg. On Ketamine-Induced Erectile Dysfunction in Rats” is well composed. The research topic is interesting. I nave only one comment: In chapter 4. Materials and Methods - chapter 4.3 and chapter 4.9 are the same. This needs to be changed.

Therefore, I suggest minor revision.

R1: Thank you for your comments, we corrected and erased that repeated point.